# Theoretical Strategy for Interface Design and Thermal Performance Prediction in Diamond/Aluminum Composite Based on Scattering-Mediated Acoustic Mismatch Model

**DOI:** 10.3390/ma16124208

**Published:** 2023-06-06

**Authors:** Zhiliang Hua, Kang Wang, Wenfang Li, Zhiyan Chen

**Affiliations:** 1School of Materials Science and Engineering, Central South University of Forestry and Technology, Changsha 410004, China; 20201100203@csuft.edu.cn; 2School of Materials Science and Engineering, Dongguan University of Technology, Dongguan 523000, China; wangk@dgut.edu.com

**Keywords:** diamond/aluminum composite, interfacial thermal conductance, interfacial structure, scattering-mediated acoustic mismatch model, thermal conductivity

## Abstract

Inserting modification layers at the diamond/Al interface is an effective technique in improving the interfacial thermal conductance (ITC) of the composite. However, few study reports the effect of interfacial structure on the thermal conductivity (TC) of diamond/Al composites at room temperature. Herein, the scattering-mediated acoustic mismatch model, suitable for evaluating the ITC at room temperature, is utilized to predict the TC performance of the diamond/Al composite. According to the practical microstructure of the composites, the reaction products at diamond/Al interface on the TC performance are concerned. Results indicate that the TC of the diamond/Al composite is dominantly affected by the thickness, the Debye temperature and the TC of the interfacial phase, meeting with multiple documented results. This work provides a method to assess the interfacial structure on the TC performance of metal matrix composite at room temperature.

## 1. Introduction

Diamond/aluminum composites have drawn much attention in microelectronics and semiconductors due to their excellent thermal conductivity (TC), tunable coefficient of thermal expansion, and light weight [1,2]. Earlier studies have shown that a well-bonded interface between diamond and aluminum is beneficial for improving the TC and interfacial thermal conductance (ITC) of diamond/Al composites [3,4,5]. However, owing to significant differences in the physicochemical properties of diamond and aluminum, diamond/Al composites exhibit poor interfacial bonding and a low thermal performance. Additionally, earlier studies have shown that the direct contact of diamond and Al at high temperatures will cause them to react to produce aluminum carbide (Al_4_C_3_) at the interface, which is beneficial for improving the interfacial bonding of composites to a certain extent [6,7,8]. Thus, a diamond/aluminum composite with a small amount of Al_4_C_3_ products demonstrated exceptional thermal conductivity, reaching a noteworthy value of 1024 W/mK [9]. However, excessive Al_4_C_3_ will inevitably lead to a significant deterioration in the ITC for diamond/Al composites due to its low intrinsic TC. Moreover, Al_4_C_3_ is prone to hydrolysis in humid conditions, which significantly limits the practical applications of diamond/Al composites.

In order to address the problem mentioned above, matrix alloying [10,11,12] and the surface metallization of diamond particles [11,12,13,14,15,16,17,18,19,20,21,22,23,24] have been widely used to generate an interfacial layer between diamond and aluminum, which is designed to prevent the formation of Al_4_C_3_ products and promote the formation of effective compounds, such as intermetallic products and carbides. Furthermore, adding an interface layer between diamond and aluminum can effectively address the inherent bonding deficiencies at the interface, and thus enhance the interfacial bonding and improve the thermal performance of diamond/Al composites. However, alloying will cause a significant deterioration in the TC of the metal matrix, resulting in a dissatisfactory TC of diamond/Al composites. On the contrary, the surface metallization of diamond particles is the more effective method for achieving a high TC of composites, which is the focus of the research. Several elements, such as W [13,14,15,16], Ti [11,12,17,18,19], Cr [20], Si [21], Zr [22], B [23], and Mo [24], have been widely used for diamond surface metallization, most of which are proposed to be effective (Figure 1). However, the introduction of an interface layer will inevitably lead to a change in the interfacial thermal resistance (ITR) of diamond/Al composites due to the layer’s inherent thermal resistance.

It is worth noting that the value of ITC plays a crucial role in determining the TC of diamond/Al composites [25]. The acoustic mismatch model (AMM) proposed by Halatnikov [26] and the diffusion mismatch model (DMM) proposed by Swartz [27] have been extensively applied in evaluating and designing the ITC of diamond/Al composites. Nevertheless, Swartz [27] and Stevens [28] pointed out that neither the AMM nor DMM considers the scattering of hot carriers at the interface between both sides of the solid phases at room temperature. As a result, neither the AMM nor DMM can accurately predict the ITC of composites because the applicable temperature was only below 50 K. To solve the problems mentioned above, Prasher and Phelan [29] proposed a new model named the scattering-mediated acoustic mismatch model (SMAMM), which was developed on the analogy of the heat transport between radiative and phonon. In this predictive model, it is assumed that there is no diffuse scattering at the interface, and the Debye velocity of phonons is taken to investigate the impact of scattering on the ITR of the composites. It is important to note that the ITR is the reciprocal of the ITC. To further validate the feasibility and applicability of the SMAMM, Battabyal et al. [30] investigated the ITR of diamond/Al composites without an interlayer based on the SMAMM at room temperature. The results showed that the predictive value as derived from the SMAMM was 4.44 × 10^−9^ m^2^·K/W, while the values derived from the AMM and the DMM were 2.42 × 10^−9^ m^2^·K/W and 2.74 × 10^−9^ m^2^·K/W, respectively. Furthermore, the resulting value was 18% lower than the nominal thermal resistance of composite, which indicates that the value of the ITR derived from the SMAMM more closely approached the practical ITR than the AMM and DMM.

Surface-metallized diamond may introduce an interface layer in the preparation of surface-metallized diamond/Al composites, leading to a change in the interfacial thermal resistance (ITR) of diamond/Al composites due to the layer’s inherent thermal resistance. The value of ITR is the critical factor in determining the ITC and TC of diamond/Al composites. Although previous studies have proven that the SMAMM has a higher accuracy when predicting the ITR of diamond/Al composites without interlayers, there are fewer studies that have predicted the thermal performance of diamond/Al composites with interfacial layers based on the SMAMM, which has drawn our attention. Therefore, the current work aims to examine the effects of various interface layers, structures, and thicknesses on the ITC and TC of diamond/Al composites based on the scattering-mediated acoustic mismatch model [29,30] (SMAMM) and the differential effective medium [31,32] (DEM), respectively, which can provide insights for the design and production of composites.

## 2. Modeling

The calculation for the interfacial thermal conductance (ITC) and thermal conductivity (TC) of a diamond/Al composite with an interfacial layer is mainly composed of four parts. The first step is to establish the simplification of the interlayer structure in diamond/Al composites, and then establish the theoretical formulas for the thermal resistance (R_b_) and ITC of composites. Lastly, the ITC and TC of the composites are calculated based on the SMAMM and the DEM model, respectively. The detailed calculation steps are shown in the following sections.

### 2.1. Simplification of the Interlayer Structure of the Diamond/Al Composite

Generally, the carbon atoms present on the surface of the diamond may react with metal-forming layers at high temperatures during the production of diamond/Al composites, leading to the formation of carbides. Additionally, the plating metal may diffuse into the aluminum matrix, resulting in the forming of intermetallic products. Thus, the composite intermediate layer was primarily composed of three phases: carbide, plated metal and intermetallic products. The thermal resistance (Rb) of diamond/Al composites with interfacial layers has several components [25]: the ITR of the diamond/carbides, the carbides/metal, the metal/intermetallics, and the intermetallics/aluminum, as well as the intrinsic resistance of the interface layers itself. A diagram illustrating the structure of the interface is shown in Figure 2.

### 2.2. Establishment of the Thermal Resistance (R_b_) and Interfacial Thermal Conductance (ITC) of Diamond/Al Composites

The *R_b_* of diamond/Al composites can be derived from the mathematical relationship between the interface temperature difference (Δ*T*) and the interface heat flow (*q*), which was proposed by Kapitza [33] based on extensive experiments. The specific mathematical relationships are as follows:(1)$$R_{b}=\frac{T_{2}-T_{1}}{q}=\frac{\Delta T}{q}$$
where *T_1_* and *T_2_* represent the temperatures of the two phases.

Owing to the similarity between the transport behavior of hot carriers and electric carriers in matter [25], the *R_b_* in the composite material can be expressed as follows:(2)$$R_{b}=\sum \left(R_{\mathrm{interface}}{+R}_{\mathrm{resistance}}\right)$$
where *R_interface_* represents the thermal resistance of the interface between two adjacent phases and *R_resistance_* represents the intrinsic resistance of the interface layer, which is determined by the TC and the thickness of the interface layer. Its expression is *R* = l/K, where l and K are the interfacial layer’s thickness and the intrinsic TC, respectively.

Thus, the theoretical expression for the *R_b_* of a diamond/Al composite with an interfacial layer is:(3)$$R_{b}{=R}_{D/C}{+R}_{C}{+R}_{C/M}{+R}_{M}{+R}_{M/I}{+R}_{I}{+R}_{I/\mathrm{Matrix}}$$
where *D*, *C*, *M*, *I*, and matrix represent the diamond, carbide, metal, intermetallic, and matrix components, respectively.

Since the *h_c_* is defined as the reciprocal of the *R_b_*, it can be expressed by the equation: $h_{c}=\frac{1}{R_{b}}$. Thus, the ITC *h_c_* of the diamond/Al composites with interface layers can be expressed as:(4)$$\frac{1}{h_{c}}=\frac{1}{h_{D-C}}+\frac{1}{h_{C-M}}+\frac{1}{h_{M-I}}+\frac{1}{h_{I-\mathrm{Matrix}}}+\frac{l_{C}}{K_{C}}+\frac{l_{M}}{K_{M}}+\frac{l_{I}}{K_{I}}$$
where *D*, *C*, *M*, *I*, and matrix have the same meanings as in Equation (3) and *l* and *K* represent the thickness and intrinsic TC of the interfacial layer, respectively.

In order to simplify the calculation, it was assumed that the metal coating did not react with the aluminum matrix, which means that no intermetallic compounds were formed at the interface. Additionally, the interfacial layers were treated as individuals and they can exist independently. As a result, the equation for the theoretical ITC of the composite can be expressed as:(5)$$\frac{1}{h}=\frac{1}{h_{D-C}}+\frac{1}{h_{C-M}}+\frac{1}{h_{M-\mathrm{matrix}}}+\frac{l_{C}}{K_{C}}+\frac{l_{M}}{K_{M}}$$
where the symbols have the same meanings as in Equation (3) and the value of *h_D-C_*, *h_C-M_*, and *h_M-matrix_* can be calculated with the scattering-mediated acoustic mismatch model (SMAMM).

### 2.3. Theoretical Calculation of the Interfacial Thermal Conductance (ITC)

Before delving into the ITC calculation, some critical assumptions were made, which are described as follows. Firstly, the phonons were assumed to be the dominant heat carriers for heat transportation through the interfaces. Secondly, the effects of other modes were considered by using the phonon velocity, and the mode conversion was ignored [29].

According to the treatise proposed by Prasher [29], the net heat flux (*q*) of two adjacent phases can be calculated by the SMAMM:(6)$$q=\frac{3}{{4\pi }^{2}}\frac{1}{V_{1}^{2}}\overset{\frac{\pi }{2}}{\underset{0}{\int }}\overset{\omega_{d<}}{\underset{0}{\int }}{h\omega }^{3}\left[\frac{1}{\exp\left(\frac{h\omega }{K_{b}T_{2}}\right)-1}-\frac{1}{\exp\left(\frac{h\omega }{K_{b}T_{1}}\right)-1}\right]\;\times \;\alpha \left(\theta ,\omega \right)\sin\left(\theta \right)\cos\left(\theta \right)d\omega d\theta$$
where *V* is the Debye speed for phonons; 1 and 2 are the non-attenuating and attenuating media, respectively; T is the temperature; θ is the phonon incidence angle; α(θ,ω) is the transitivity of phonons; and *ω_d_*_<_ is the smaller Debye frequency between the two phases, which affects the scattering mismatch process of the phonons. *ω_d_*_<_ can be calculated by $\omega =\frac{K_{b}\theta_{d}}{h}$, where h and K_b_ are the approximate Planck constant (1.055 × 10^−34^ J·S) and the Boltzmann constant (1.3806 × 10^−23^ J·K^−1^), respectively. Since $\frac{K_{b}}{h}$ is a constant value, the scattering mismatch process of phonons is primarily determined by the magnitude of the Debye temperature (θ_d_).

Here, the materials with a higher Debye temperature, which were non-attenuating media, had a higher phonon mean free range and a larger sound velocity [29], while the others were attenuating media. Since the critical angle θ occurred on the side of the attenuating media, the calculation of the interfacial heat flux q value was carried out using the non-damping media. Therefore, there was no need to worry about the influence of the critical angle in the calculation, and the important factor was to determine the rate of phonon transmission α.

To solve the problem mentioned above, a formula was derived to describe the transmissivity (α) for the angle (θ_1_) of incidence, which can be expressed as follows:(7)$$\alpha \left(\theta_{1}\right)=\frac{{4P}_{21}\cos\left(\theta_{1}\right)\sqrt{1-{\left[\frac{V_{2}}{V_{1}}\sin\left(\theta_{1}\right)\right]}^{2}}}{{\left[\cos\left(\theta_{1}\right){+P}_{21}\sqrt{1-{\left[\frac{V_{2}}{V_{1}}\sin\left(\theta_{1}\right)\right]}^{2}}\right]}^{2}}$$
where V is the Debye speed for phonons; 1 and 2 represent the non-attenuating medium and the attenuating medium, respectively; θ is the phonon incidence angle; θ_d_ is the Debye temperature; M is the atomic mass; and P_21_ = 1 [29].

Since the net heat flux was calculated from Equation (6), the ITC of two adjacent materials can be calculated from the interface temperature difference and the net heat flux:(8)$$h_{c}=\frac{q}{\Delta T}$$
where a temperature difference of 0.2 K was chosen for the calculation of h_c_ (∆T = 0.2 K).

Thus, the values of h_D-C_, h_C-M_, and h_M-matrix_ can be calculated simultaneously by Equations (6)–(8). The physical parameters of various materials used for the calculation in this work are listed in Table 1.

To verify the reliability of the SMAMM, the calculation for the heat flux (q) of a diamond/Al composite without an interlayer was conducted. Firstly, the crystallographic parameters of the interface and the transmissivity for the angle were imported into the SMAMM, and then a mathematical analysis of the heat flux values was performed through double integration. The output result is presented in Figure 3, which coincided with the outcomes published by Battabyal [30].

### 2.4. Theoretical Calculation of Thermal Conductivity (TC)

As a significant model for predicting the TC (*K_c_*) of diamond/Al composites, the differential effective medium [31,32] (DEM) model was adopted in this work. Its expression is:(9)$$\left(1-V_{r}\right){\left(\frac{K_{C}}{K_{m}}\right)}^{\frac{1}{3}}=\frac{K_{r}^{\mathrm{eff}}-K_{c}}{K_{r}^{\mathrm{eff}}-K_{m}}\;$$
(10)$$K_{r}^{\mathrm{eff}}=\frac{K_{r}}{1+\frac{K_{r}}{h_{c}a}}$$
where *K_c_* and *K_m_* are the TC values of the composite and metal matrix, respectively; *Vr* is the volume fraction of the reinforcing phase; *Kr* is the TC of the reinforcing phase; *h_c_* is the ITC of composites; and a is the average size of the reinforcing phase. Here, the diamond particle size and a volume of 150 μm and 50% were taken to calculate the ITC and TC of diamond/Al with different interfacial layers.

## 3. Results and Discussion

According to the mention in Section 2, metal elements may react with carbon atoms to produce carbides at the interface at high temperatures, which has been observed in the formation of composites. This means that there are three interfacial structures inside the composites, namely metals, carbides, or both metals and carbides. Therefore, an evaluation of various structures on the ITC and TC of diamond/aluminum composites was necessary, and it was computed as shown in the following sections. Specifically, the effect of various carbide-forming metals, their corresponding carbides, and the carbide transformation on the ITC and TC of diamond/Al composites was evaluated based on the SMAMM and the DEM model, respectively, which are presented in the Figure as shown below.

### 3.1. Effect of Carbide-Forming Metal Layers

As shown in Figure 4, the ITC and TC of diamond/Al composites with various interfacial layers decreased with a power-law function when the layer thickness increased. The calculation result showed that the diamond/Al composites with B, Si, and Cr layers with a thickness of 50 nm effectively improved the ITC and TC, and there was a small difference in improving the TC. However, the Mo, Ti, W, and Zr layers were not favorable for improving the ITC and TC of the composites, regardless of their thickness. By constructing different types of layers between the diamond and Al matrix, the variation in the ITC and TC of the composites can be explained by the differences in the Debye temperature and the phonon velocity of the interface layers based on the SMAMM.

It was reported that there is a significant difference in the Debye temperature between diamond and aluminum, leading to a significant difference in the cut-off frequencies of the two phases. Specifically, aluminum has a cut-off frequency between 2~5 × 10^13^ s^−1^, while diamond has a cut-off frequency of 2.87 × 10^14^ s^−1^. This means that only the phonons of Al, whose phonon density of states matches that of diamond phonons in the Al frequency range, can effectively pass through the interface, and thus interact with diamond phonons, while the other phonons will be either scattered or reflected [30]. Thus, the phonon–phonon transfer efficiency between aluminum and diamond is low, resulting in a low interface heat flux and an unfavorable thermal performance. However, the interface layers with a high Debye temperature can establish an intermediate stage for phonon–phonon transmission between diamond and Al, which can effectively restrain the scattering phenomenon of phonons, and thus improve the phonon–phonon transfer efficiency and interface heat flux density (Equation (6)) of the diamond/Al composites. Moreover, materials with higher Debye temperatures have a more significant phonon velocity. The dissimilarity in the phonon velocity between diamond and aluminum leads to phonon scattering at the interface. However, an interfacial layer with a more significant phonon velocity creates a helpful gradient of phonons at the interface, effectively suppressing the phonon scattering at the interface.

Therefore, B, Si, and Cr layers positively impacted the ITC and TC of the composite at a specific layer thickness due to their more significant phonon velocities and Debye temperatures, while Ti, Zr, W, and Mo layers did not (Figure 5). Specially, the Mo layer impeded heat transfer through the interface of the diamond/Al composite because the Debye phonon velocity was lower than that of the Al (6250 and 6402 m/s, respectively), despite its higher Debye temperature. It is worth noting that the phonon velocity in this region is the keynote of the ITC and TC of diamond/Al composites. As shown in Table 1, the B and Si layers had higher phonon velocities and Debye temperatures compared to those of Cr, which is beneficial for improving the ITC and TC of a diamond/Al composite at smaller layer thicknesses. For instance, with a 50 nm-thick layer, the TC and ITC of diamond/Al composites with B and Si layers were 847.3 W/m K and 40.6 × 10^7^ W/m^2^·K for B and 846.5 W/m^2^·K and 37.1 × 10^7^ W/m^2^·K for Si, respectively, which were higher than the 841.1 W/m K and 24.47 × 10^7^ W/m^2^·K of the diamond/Al composite without an intermediate layer. However, the TC and ITC of the diamond/Al composite with a Cr layer were 843.5 W/m K and 27.3 × 10^7^ W/m^2^·K, respectively, which were closer to the values of pure diamond/Al. Furthermore, B was the most prospective interlayer for improving the TC and ITC of composites with a layer thickness of 50 nm. However, considering the intrinsic TC of the interfacial layer, Si was the most significant layer for improving the TC and ITC of diamond/Al composites when the layer thickness was more than 50 nm.

As shown in Figure 4b, the ITC and TC of diamond/Al with Ti, Zr, and B layers decreased dramatically as the interfacial layer thickness increased, whereas those of composites with Mo, Si, Cr, and W layers decreased more gradually due to their larger intrinsic TC. It was noted that the interface layers with lower intrinsic TC values had higher intrinsic thermal resistance values. For example, with the same layer thickness of 500 nm, the intrinsic thermal resistance value of the Ti layer was 22.7 × 10^−9^ m^2^·K/W, much higher than the 3.96 × 10^−9^ m^2^·K/W of the Si layer. Consequently, the TC and ITC of diamond/Al composites with Ti, Zr, and B layers showed a significant decrease as the interface layer thickness increased, while the others did not. 

In summary, the Cr, B, and Si interfacial layers within a thickness of 50 nm played a positive role in improving the TC and ITC of diamond/Al composites, especially the B layer. However, when the thickness was more than 50 nm, Si was the most significant layer for improving the TC and ITC of diamond/Al composites due to its higher intrinsic TC.

### 3.2. Effect of Carbide Layers

The introduction of carbides has proved to be double-edged. On the one hand, introducing carbide layers in diamond/Al composites results in the formation of a gradient in the sound velocity between the diamond and aluminum, which plays a positive role in improving the phonon–phonon (ph–ph) coupling, thus leading to an increasing trend in the ITC and TC of the composite. On the other hand, it is essential to note that the carbide layer’s thermal resistance plays a negative role in improving the TC of the composite. As shown in Figure 6, all carbide interfacial layers except Mo_2_C were beneficial for improving the TC and ITC of a diamond/Al composite when the layer thickness was no more than 50 nm. Additionally, there was only a small difference in the improvement of the TC of composites. Moreover, the B_4_C and SiC layers had significant advantages in terms of improving the TC of composites due to their larger Debye temperatures and phonon velocities (Figure 7), especially when the layer thickness was more than 100 nm. For the diamond/Al composites with a Mo_2_C interfacial layer, it is worth noting that the Mo2C layer had a lower phonon velocity compared to the aluminum matrix (Mo^2^C: 6257 m/s; Al: 6402 m/s), making it impossible to form a phonon velocity gradient in the composite. As a result, the Mo_2_C layer played a negative role in improving the TC and ITC of the composite, regardless of the layer thickness. 

As can also be observed from Figure 6, the ITC and TC of the composites showed a decreasing trend as the carbide layer thickness gradually increased. In particular, the TC values of the Cr_7_C_3_, Cr_3_C_2_, TiC, and ZrC layers displayed a significant downward trend, while those of the SiC, WC, and B_4_C layers showed a significant decrease. The difference in the trend can be explained by the different intrinsic TC values of the carbide layers. When the composites with 500 nm-thick layers were prepared, the intrinsic thermal resistance values of the ZrC and SiC interface layers were 23.81 × 10^−9^ m^2^·K/W and 2.79 × 10^−9^ m^2^·K/W, respectively. It can be seen that the introduction of a ZrC layer resulted in a higher interfacial thermal resistance of the diamond/Al composites compared to SiC. Therefore, with the introduction of Cr_7_C_3_, Cr_3_C_2_, TiC, and ZrC carbide layers, the effect of layer thickness on the TC should be prioritized, while the SiC, WC, and B_4_C carbide layers may be better suited to improving the thermal performance of diamond/Al composites due to their high thermal conductivity (Table 1). However, considering the intrinsic resistance of the interface layer, the SiC layer is the most advantageous for improving the thermal conductivity of diamond/Al composites.

### 3.3. Effect of Carbide Transformation

The metal layer on diamond’s surface may partially convert to the carbide layer at high temperatures. In other words, the interfacial structure of the composite consists of both metal and carbide layers. Therefore, it is essential to investigate the impact of the carbide percentage in the intermediate layer on the TC of diamond/Al composites. As presented in Figure 4, the TC and ITC of composites with different interface layers of 250 nm exhibited significant changes at this critical turning point. Thus, the theoretical calculations were conducted specifically for a layer thickness of 250 nm, aiming to investigate the effect of carbide transformation on the ITC and TC of the composites. As shown in Figure 8, the results revealed that, as the proportion of interfacial carbides increased, the ITC and TC of the composites with Cr–Cr_7_C_3_, Cr–Cr_3_C_2_, and Mo–Mo_2_C layers first increased and then dropped, but overall displayed a downward trend. Conversely, the ITC and TC of the composites with W–WC, Zr–ZrC, Si–SiC, Ti–TiC, and B–B_4_C layers showed slight changes at certain carbide conversion stages, and the overall curve trend was upwards. Notably, the ITC and TC of the diamond/Al composites with a Si–SiC layer reached the highest values of 4.68 × 10^8^ W/m^2^·K and 848.5 W/m·K, respectively.

The decreasing trend observed in the ITC and TC curves of the composites with Cr–Cr_7_C_3_, Cr–Cr_3_C_2_, and Mo–Mo_2_C layers in Figure 8 can be attributed to the formation of carbides. Although the generative carbide layer can reduce the diffusion of thermal carriers from interfaces because its higher phonon velocity and Debye temperature, the TC of the generative carbide layer after transformation was significantly lower than that of the metal. The trend in the curves was primarily governed by the TC of the transformation substance. However, when the carbide proportion was up to 90–100%, the ITC and TC of the composites exhibited a slight rise. This could be explained by the difference in the Debye temperature and phonon velocity. The diamond/carbide/Al interface was better at reducing the acoustic mismatch and scattering of heat carriers compared to the diamond/metal/Al interface. For example, when the Cr element was converted into Cr_7_C_3_ at high temperatures, there was a notable variation in the thermal conductivity between the two states. The TC of pure Cr was approximately 90 W/m·K, while that of Cr_7_C_3_ was only about 19.1 W/m·K. Thus, the intrinsic thermal resistance of the pure Cr interface layer was 2.77 × 10^−9^ m^2^·K/W. However, when the proportion of carbides was up to 10%, the intrinsic thermal resistance of the Cr–Cr_7_C_3_ layer increased to a value of 3.809 × 10^−9^ m^2^·K/W, which was higher than that of the pure Cr interface layer. The composite experienced an increase in thermal resistance with an increase in the amount of Cr_7_C_3_ produced, leading to a noticeable decrease in the TC. As soon as the transformation of the Cr element into Cr_7_C_3_ was complete, the thermal flux at the composite material interface experienced an upsurge due to the excellent properties of the Cr_7_C_3_ interface layer. Specifically, the higher Debye temperature of 646 K and the higher phonon velocity of 8218 m/s of the Cr_7_C_3_ material contributed to the increase in heat flux, which led to a subsequent improvement in the thermal performance of the composite.

The ITC and TC curves of the interface layers (W–WC, Zr–ZrC, Si–SiC, Ti–TiC, and B–B_4_C) showed a general upward trend, probably because the carbides forming at the interface had a higher phonon velocity, a higher Debye temperature, and a higher intrinsic TC than those of the elemental layers, thus positively affecting the ITC and TC of the composites. For instance, when Zr was transformed into ZrC at high temperatures, the ITC and TC of the composite exhibited a significant upward trend with a carbide conversion range of 0 to 10%. This can be attributed to the influence of the more significant phonon velocity and Debye temperature of ZrC. As the carbonization rate increased from 0 to 10%, the ITC and TC of the composite material increased from 768.8 W/m K and 4.14 × 10^7^ W/m^2^·K to 806.3 W/m K and 7.54 × 10^7^ W/m^2^·K, respectively. However, there was no significant change in the ITC and TC of the composites between a carbide content of 10–90% in the middle layer, probably because the difference in the intrinsic TC between Zr and ZrC is not substantial (21 W/m·K).

### 3.4. Suggested Optimization Interface and a Comparison with References

Figure 9a is a summary of the calculation for the interfacial thermal conductance (ITC) and thermal conductivity (TC) of diamond/Al composites with various interface layers. As shown in Figure 9a, it should be noted that the interfacial bonding of the composite was idealized when the interface layer thickness was 1 nm, without considering the intrinsic thermal resistance of the layer itself. As for the case of introducing the interface layers, the Cr, B, Si, SiC, B_4_C, TiC, WC, Cr_7_C_3_, Cr_3_C_2_, and ZrC layers with a nanoscale thickness positively improved the thermal performance of the diamond/Al. However, when considering the intrinsic TC of the interface layers, SiC was the most promising interfacial layer for improving the ITC and TC of diamond/Al composites. Figure 9b shows the TC of diamond/Al composites with various interfacial layers chosen from different references and theoretical calculations. However, the TC of diamond/Al composites with different layers, calculated theoretically, was higher than that of the composites chosen from different references, which was mainly due to the poor interfacial bonding of the composites, the formation of an intermetallic compound, and the negative effect of the solute dissolution in the Al matrix. All of these factors significantly affected the TC of the diamond/Al composite.

Here, it is worth noting that the predictive model was effective for evaluating the thermal performance of diamond/Al composites with a perfect interfacial bonding, while the thermal performance of diamond/Al prepared by different methods may be affected by poor interfacial bonding. Additionally, intermetallic products at the interface had a significant impact on the interfacial thermal resistance (ITR) of the diamond/Al composites. Significantly, the ITR was the critical factor in determining the TC of diamond/Al composites. Tan et al. [25] reported that intermetallic products possess a large sound velocity, which plays a positive role in improving the TC and interfacial bonding of the composites. However, owing to the excessive intermetallic products with a low intrinsic TC, an increase in the ITR of the composite is harmful to the TC and ITC of the composite. For instance, Battabyal et al. [30] revealed that the value of the experimental ITR for the pure diamond/Al composite was 5.43 × 10^−9^ m^2^·K/W, while the ITR derived by the predictive model was 4.44 × 10^−9^ m^2^·K/W. However, it should be noted that Al_4_C_3_ is prone to be produced at the interface at high temperatures in diamond/Al composites. By assuming that the layer thickness of Al_4_C_3_ is 1 nm and 500 nm, the theoretical ITR of the diamond/Al_4_C_3_/Al composite was calculated to be 1.08 × 10^−9^ m^2^·K/W and 4.65 × 10^−9^ m^2^·K/W, respectively (Table 2). According to the results mentioned above, it can be clearly observed that the Al_4_C_3_ interfacial layer with a thickness of 1 nm was beneficial for reducing the ITR of the composite. As the Al_4_C_3_ thickness increased to 500 nm, the ITR of the composite showed an upward trend, which was much closer to the experimental results. Although the intermetallic products had a significant effect on the ITR of the composites, there has been no systematic study on their effect in surface-metallized diamond/Al composites. Moreover, the solubility of the metal layer in the Al matrix also significantly leads to deteriorate the TC of diamond/Al composites, which has been verified in previous studies.

Therefore, it is essential to evaluate the effect of interfacial bonding, intermetallic formation, and solute dissolution in the Al matrix on the TC of diamond/Al composites, which is the main focus discussed in the future.

## 4. Conclusions

In this work, a multi-layer interface model and the SMAMM were established to evaluate the effect of various interface layers, structures, and thicknesses on the interfacial thermal conductance (ITC) and thermal conductivity (TC) of diamond/Al composites. Furthermore, the influence of various common elements (such as Cr, W, Si, Mo, Ti, Zr, and B) and their corresponding carbide layers on the TC and ITC of diamond/Al composites was considered. The results showed that an interfacial layer with a nanoscale thickness, a high intrinsic TC, a high phonon velocity, and a high Debye temperature was beneficial for improving the thermal performance of diamond/Al composites. The detailed conclusions for diamond/Al composites with interfacial layers are listed as follows:(1)The TC and ITC of diamond/Al composites with different interface layers showed a decrease when the layer thickness increased, especially for interface layers with a low intrinsic TC, a low Debye temperature, and a low phonon velocity, i.e., Mo, Ti, W, Zr, ZrC, TiC, Mo_2_C, Cr_3_C_2_, and Cr_7_C_3_, while Cr, B, Si, WC, SiC, and B_4_C layers were favorable for achieving a desirable TC.(2)Carbide-forming metals, i.e., Cr, B, and Si, served as the optimal interfacial elements for improving the thermal performance of diamond/Al composites. However, when considering the effect of carbide transformation, Si and B served as the optimal interfacial elements for improving the thermal performance of diamond/Al composites.(3)Among all the interface layers, with its high intrinsic TC, high phonon velocity, and high Debye temperature, SiC was the most promising interfacial layer for achieving a higher TC in the composites.

## Figures and Tables

**Figure 1 materials-16-04208-f001:**
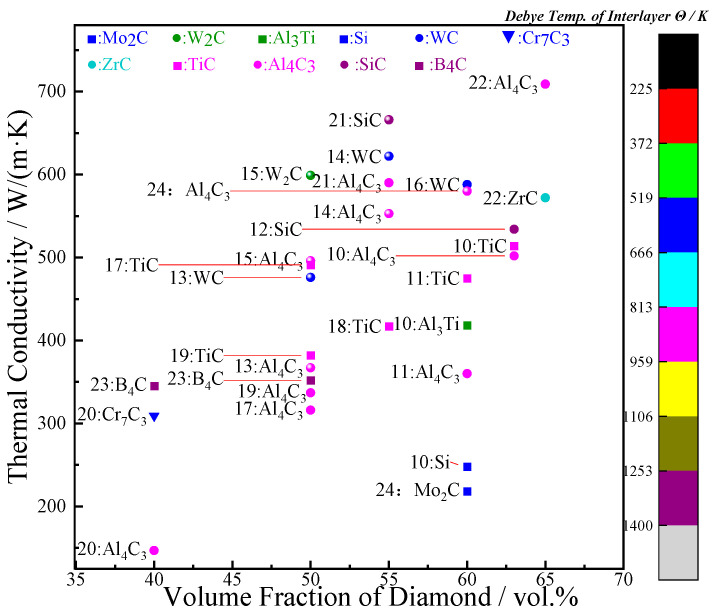
The diagram of diamond/Al composites with various interfacial layers in thermal conductivity.

**Figure 2 materials-16-04208-f002:**
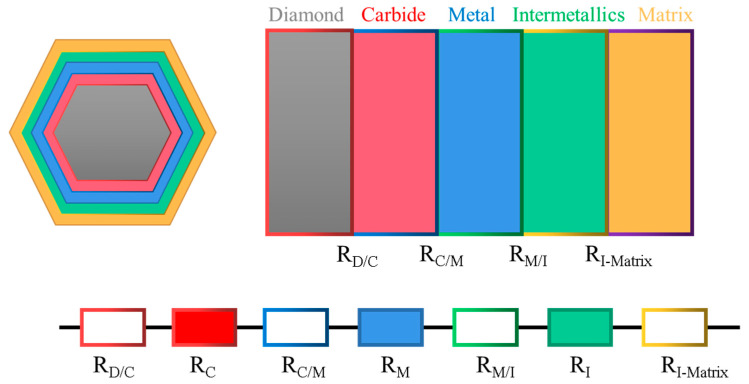
Schematic diagrams of the diamond/aluminum composites with an interfacial layer.

**Figure 3 materials-16-04208-f003:**
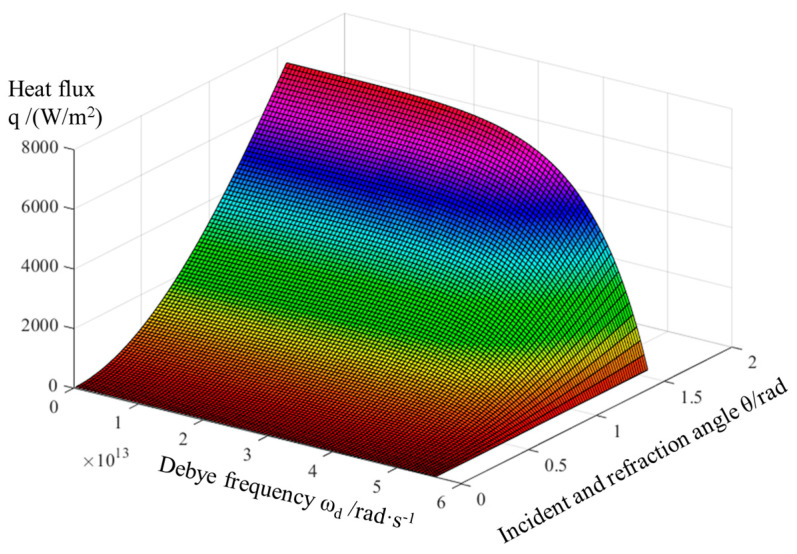
The heat flux of diamond/Al (perfect interface) based on the SMAMM.

**Figure 4 materials-16-04208-f004:**
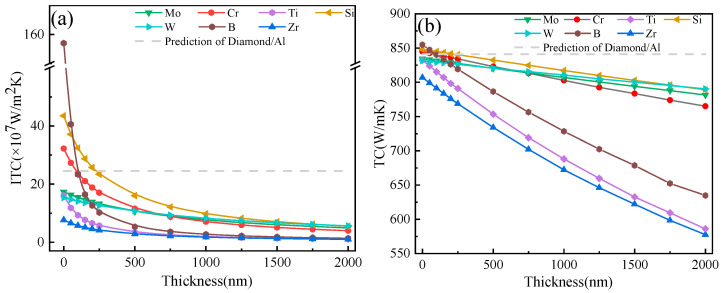
The ITC (**a**) and TC (**b**) of diamond/Al composites with different carbide-forming metal layers.

**Figure 5 materials-16-04208-f005:**
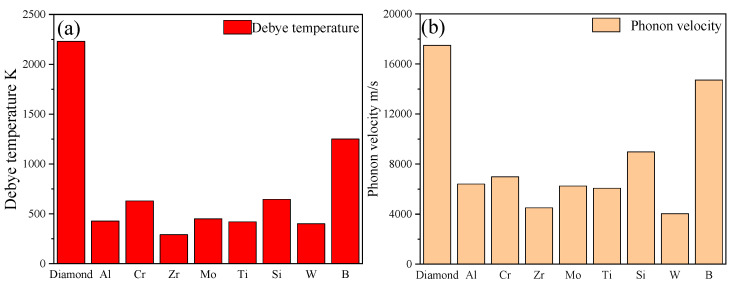
Comparisons of Debye temperature and phonon velocity in different materials: (**a**) Debye temperature and (**b**) phonon velocity.

**Figure 6 materials-16-04208-f006:**
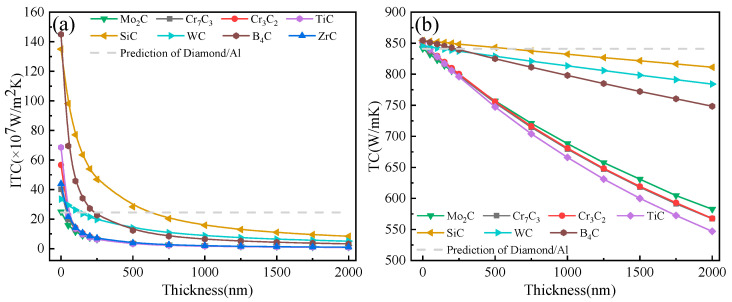
The ITC (**a**) and TC (**b**) of diamond/ Al composites with different carbide layers.

**Figure 7 materials-16-04208-f007:**
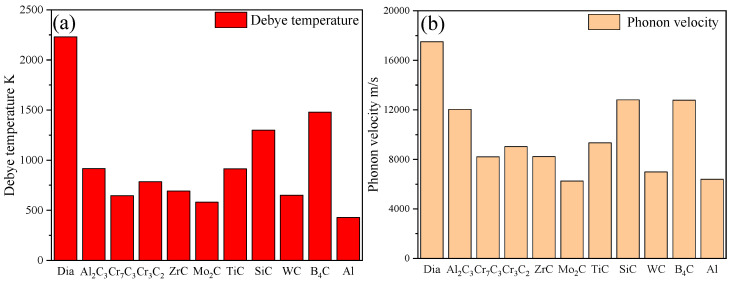
Comparisons of Debye temperature and phonon velocity in different carbides: (**a**) Debye temperature and (**b**) phonon velocity.

**Figure 8 materials-16-04208-f008:**
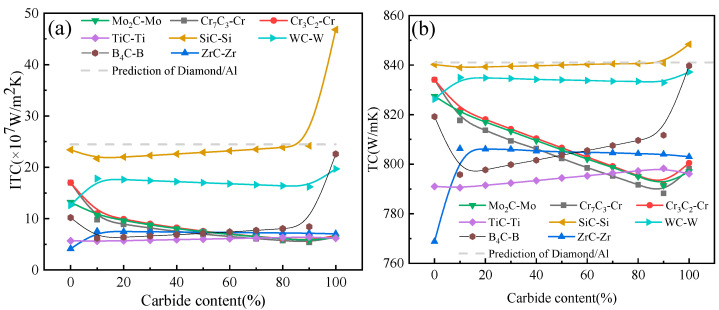
The ITC (**a**) and TC (**b**) of diamond/Al composites with 250 nm-thick interface layers but a varying carbide content.

**Figure 9 materials-16-04208-f009:**
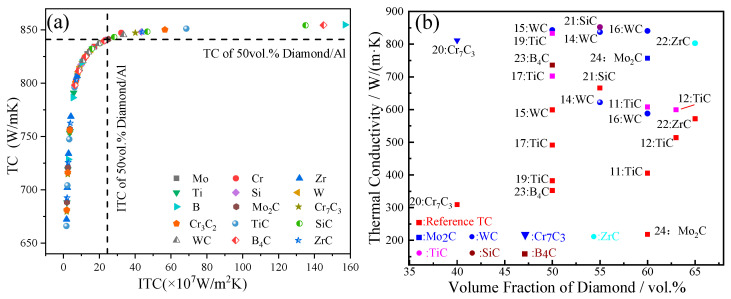
The thermal performance of diamond/Al composites with: (**a**) different layer thicknesses of 0.01–1.0 μm and (**b**) various interface layers chosen from different references and theoretical calculations.

**Table 1 materials-16-04208-t001:** The physical parameters of optimized species of a diamond/Al composite.

Materials	Density ρ/(Kg·m^3^)	Thermal Conductivity K(W/m·k)	Specific Heat C(J/Kg·k)	Phonon Velocity V/(m·s^−1^)	Debye Temperature Θd/(K)	Debye Frequency ωd/(rad·s^−1^)
Diamond	3520	1800	512	17,500 [27]	2230	29.2 × 10^13^
Al	2700	237	895	6402	428	5.60 × 10^13^
Cr	7190	90	446	6980 [25]	630	8.24 × 10^13^
Zr	6510	22.6	270	4502	291	3.80 × 10^13^
Mo	10,200	138	248	6250	450	5.89 × 10^13^
Ti	4540	22	522	6070 [34]	420	5.50 × 10^13^
Si	2330	126	703	8970	645	8.44 × 10^13^
W	19,320	178	133	4029	400	5.23 × 10^13^
B	2080	27.4	1026	14,719	1250	16.36 × 10^13^
Al_4_C_3_	2360	140	800	12,038	916 [35]	12.0 × 10^13^
Cr_7_C_3_	6920	19.1	543	8218 [36]	646 [36]	8.45 × 10^13^
Cr_3_C_2_	6680	19	456	9033 [35]	785	10.27 × 10^13^
ZrC	6730	21	364	8228 [35]	691.3 [37]	9.05 × 10^13^
Mo_2_C	9000	21	347	6257 [34]	580.9 [38]	7.60 × 10^13^
TiC	4930	17	569	9330 [34]	915	11.97 × 10^13^
SiC	3100	179	678	12,810 [34]	1300 [39]	17.03 × 10^13^
WC	14,900	120	203	6985 [35]	650.4 [40]	8.51 × 10^13^
B_4_C	2520	67	583	12,781	1480 [41]	19.37 × 10^13^

**Table 2 materials-16-04208-t002:** The ITR of diamond/Al composite gathered from references and theoretical calculations by SMAMM.

Materials	Diamond/Al (Experimental)	Diamond/Al	Diamond/Al_4_C_3_/Al (d = 0 nm)	Diamond/Al_4_C_3_/Al (d = 500 nm)
ITR (m^2^K/W)	5.43 × 10^−9^	4.44 × 10^−9^	1.08 × 10^−9^	4.65 × 10^−9^

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
