# Peer review of "Theoretical Strategy for Interface Design and Thermal Performance Prediction in Diamond/Aluminum Composite Based on Scattering-Mediated Acoustic Mismatch Model"

_materials, 2023, doi:10.3390/ma16124208_

Round 1

Reviewer 1 Report

The manuscript titled “Theoretical strategy for interface design and thermal performance prediction in diamond/aluminum composite based on 3 scattering-mediated acoustic mismatch model” is very well written. Only a few minor changes are required as listed below:

1.       Abstract should contain the specific output of the manuscript only. So authors may rewrite it

2.       The idea of doing the research is slightly not clear in the introduction section. Authors are advised to reframe the last paragraph of the introduction section.

3.       All grammatical errors and written English must be checked once again before the final submission.

4.       Conclusion section must be shortened and should represent the gist of the article. 

Reviewer 2 Report

1.       Summary, strengths, weaknesses, overall contribution           

Summary: In this paper, the multi-layer interface model and SMAMM model were established to evaluate the effect of various interface layers, structures, and thicknesses on the interfacial thermal conductance (ITC) and thermal conductivity (TC) of Diamond/Al composites

General strengths: From the scientific point of view the paper is good. The important problem and material are studied.   

General weaknesses: There is no discussion about the potential influence of the layers on the interfacial bonding strength.

The paper should be accepted if the authors will refer to the following remarks and do the necessary corrections, which would significantly improve the paper:

2.                   Minor comments

- In the introduction and discussion the Authors should also include at least estimation of the influence of the interfacial layer on the interfacial bonding strength and overall mechanical properties of the composites. The following literature can be useful: 10.3390/ma14123181; 10.1016/j.apsusc.2016.12.130;

- The discussion about the accuracy of the model or some comparison with experimental results should be provided.

Reviewer 3 Report

This is a manuscript on an interesting topic, well written with a lot of results. A few improvements are needed:

1) Authors need to explain the novelty of this work.

2) Author are kindly requested to compare their findings with literature.

3) Fig9 is very small, expanding it to make it easier for the readers.

Minor improvements required.
